# Understanding Complex Interplay among Different Instabilities in Multiferroic BiMn_7_O_12_ Using ^57^Fe Probe Mössbauer Spectroscopy

**DOI:** 10.3390/ijms25031437

**Published:** 2024-01-24

**Authors:** Iana S. Soboleva, Vladimir I. Nitsenko, Alexey V. Sobolev, Maria N. Smirnova, Alexei A. Belik, Igor A. Presniakov

**Affiliations:** 1Department of Chemistry, Lomonosov Moscow State University, Moscow 119991, Russia; janglaz@bk.ru (I.S.S.); nvova.chem@gmail.com (V.I.N.); ipresniakov1969@mail.ru (I.A.P.); 2Department of Chemistry, MSU-BIT University, Shenzhen 517182, China; 3Kurnakov Institute of General and Inorganic Chemistry of Russian Academy of Sciences (RAS), Moscow 119991, Russia; 4Research Center for Materials Nanoarchitectonics (MANA), National Institute for Materials Science (NIMS), Namiki 1-1, Tsukuba 305-0044, Ibaraki, Japan; alexei.belik@nims.go.jp

**Keywords:** manganites, Mössbauer spectroscopy, multiferroics, Jahn–Teller effect

## Abstract

Here, we report the results of a Mössbauer study on hyperfine electrical and magnetic interactions in quadruple perovskite BiMn_7_O_12_ doped with ^57^Fe probes. Measurements were performed in the temperature range of 10 K < *T* < 670 K, wherein BiMn_6.96_^57^Fe_0.04_O_12_ undergoes a cascade of structural (*T*_1_ ≈ 590 K, *T*_2_ ≈ 442 K, and *T*_3_ ≈ 240 K) and magnetic (*T*_N1_ ≈ 57 K, *T*_N2_ ≈ 50 K, and *T*_N3_ ≈ 24 K) phase transitions. The analysis of the electric field gradient (EFG) parameters, including the dipole contribution from Bi^3+^ ions, confirmed the presence of the local dipole moments *p*_Bi_, which are randomly oriented in the paraelectric cubic phase (*T* > *T*_1_). The unusual behavior of the parameters of hyperfine interactions between *T*_1_ and *T*_2_ was attributed to the dynamic Jahn–Teller effect that leads to the softening of the orbital mode of Mn^3+^ ions. The parameters of the hyperfine interactions of ^57^Fe in the phases with non-zero spontaneous electrical polarization (*P*_s_), including the *P*1 ↔ *Im* transition at *T*_3_, were analyzed. On the basis of the structural data and the quadrupole splitting Δ(*T*) derived from the ^57^Fe Mössbauer spectra, the algorithm, based on the Born effective charge model, is proposed to describe *P*_s_(*T*) dependence. The *P*_s_(*T*) dependence around the *Im* ↔ *I*2/*m* phase transition at *T*_2_ is analyzed using the effective field approach. Possible reasons for the complex relaxation behavior of the spectra in the magnetically ordered states (*T* < *T*_N1_) are also discussed.

## 1. Introduction

The variety of structural and magnetic phase transitions in the so-called quadruple perovskite BiMn_7_O_12_ [1,2,3,4] and its solid solutions, such as BiMn_7−x_Cu_x_O_12_ (0 < *x* ≤ 1.1) [5,6], has attracted strong interest from researchers. Numerous transitions of different origins are associated with the presence of Mn^3+^ and Bi^3+^ cations in the crystal lattice of these oxides, which promotes structural instability [7,8,9,10]. High-spin Jahn–Teller (JT) cations Mn^3+^(*d*^4^) in the non-distorted octahedral oxygen surrounding possess an energetically degenerate configuration, *e_g_*^1^, which provokes, along with a local distortion of polyhedra (MnO_6_), a cooperative interaction in which the JT centers Mn^3+^ proper, often called orbital ordering [7,8,11,12]. The distortion driven by easily polarized Bi^3+^ cations, containing the 6*s*^2^ lone electron pair, results in off-centric cation displacements and the formation of local electric dipoles, which are responsible for the ferroelectric properties of many Bi-containing perovskites [7,10].

Non-zero magnetization (*M*) co-existing with electrical polarization (*P*_s_) is characteristic of multiferroics, which can be grouped into two types [13]. In the first group (type-I multiferroics), *M* and *P*_s_ are independent of each other, i.e., magnetism and ferroelectricity have different origins. In the second group (type-II multiferroics), *M* and *P*_s_ demonstrate a strong mutual influence, i.e., magnetism induces ferroelectricity [13]. Such a magneto-electric coupling strongly correlates with local distortions, and, thus, the roles of local polar and magnetic clusters must be studied. Therefore, these compounds are increasingly studied not only by using diffraction and magnetic methods but also using local nuclear resonance techniques, such as NMR [14,15,16,17,18,19,20,21], NQR [22,23], muon spectroscopy [24,25,26], the spectroscopy of perturbed angular γ–γ correlations [27,28], and Mössbauer spectroscopy [9,29,30]. The temperature dependences of hyperfine magnetic fields (*B*_hf_) and principal components {*V*_ii_}_X,Y,Z_ of the tensor of the electrical field gradient (EFG) gained using these methods reproduce the corresponding dependences *M*(*T*) and *P*_s_(*T*). Meanwhile, the relationship *B*_hf_ = α*M* is usually linear for ^57^Fe, ^55^Mn, and ^53^Cr nuclei, and the dependencies of the EFG tensor parameters *V*_ii_ = f(*P*_s_) and *η* = f′(*P*_s_) (where *η* = (*V*_XX_ − *V*_YY_)/*V*_ZZ_ is the asymmetry parameter) demonstrate nontrivial behavior. In some works, quadratic dependences *V*_ii_ = *a* + *bP*_s_^2^ and *η* ∝ *P*_s_^2^ are used for approximation [31,32,33]. However, such an approach is rather formal and does not allow for associating the hyperfine parameters of different resonant nuclei with the structural data and physical characteristics of the compounds under study. 

In the present work, we present Mössbauer-based research on the hyperfine interactions of ^57^Fe probes in perovskite BiMn_7_O_12_. This manganite demonstrates spontaneous electrical polarization in the temperature range of *T* < *T*_C_ ≈ 440 K, whereas, at lower temperatures (*T* < *T*_N1_ ≈ 59 K), it acquires a magnetically ordered state and multiferroic properties [2]. In contrast to perovskites *AB*O_3_, in their “quadruple” analogs (*A*′*A*″_3_)*B*_4_O_12_, the sublattice *A* is divided into two sublattices formed by cations *A*′ with a high coordination number (CN = 8–12) and by using JT cations *A*″ = Cu^2+^ and Mn^3+^, which are located in the square planar oxygen coordination [34]. In the case of (Bi^3+^Mn^3+^_3_)[Mn^3+^_4_]O_12_, these sublattices consist of the cations *A*′ = Bi^3+^ and *A*″ = Mn^3+^, whereas the sublattice with a distorted octahedral oxygen coordination (*B*) consists of the JT cations Mn^3+^ that directly initiate orbital ordering (cooperative JT effect) [8]. A combined effect of two cations (Mn^3+^)*_B_* and (Bi^3+^)*_A_*_′_, of which the latter contains the stereochemically active lone-pair electrons [7,9,10], results in a whole cascade of structural and magnetic phase transitions in BiMn_7_O_12_ (Figure 1) [3]. However, the mechanisms and driving forces of these phase transitions are still widely discussed despite abundant experimental data and theoretical studies [35]. 

According to the results of the earlier Mössbauer studies of perovskites *A*Mn_6.96_^57^Fe_0.04_O_12_ (*A* = Ca, Sr, Cd, Pb) [35,36,37], the ^57^Fe probes are localized in the structure solely in the formal oxidation state “3+”, substituting manganese cations only in octahedral sublattice. Moreover, experimental and theoretical studies show that the hyperfine parameters of the ^57^Fe spectra reflect the features of the local crystal structure of this class of manganites. It is essential to highlight that the studies involving macroscopic diagnostic methods found the influence of the ^57^Fe probes on physical parameters to be insignificant, as well as the patterns of the structural and magnetic transitions in these oxides. Thus, utilizing Mössbauer spectroscopy to probe a more complicated BiMn_7_O_12_ system is justified by the current experimental data, and the successful application of this technique is used to study other isostructural compounds of the *A*Mn_7_O_12_ family.

Our work aims to qualitatively obtain new information about the local structure of BiMn_7_O_12_ and outline the features of the structural, electrical, and magnetic phase transitions. We describe a close interplay between the orbital and spin degrees of freedom, which is characteristic of systems with a strong electron correlation. 

The Results and Discussion section of the manuscript is divided into several parts. The first part is devoted to analyzing the effect of ^57^Fe probes on structural (*T*_1_, *T*_2_, and *T*_3_) and magnetic (*T*_N1_ and *T*_N2_) transitions in BiMn_7_O_12_ (Figure 1). Based on theoretical calculations of the EFG parameters in the paraelectric range *T* > *T*_1_, the second part discusses the crystal chemistry of Bi^3+^ cations and their influence on the transition of the bismuth sublattice into the ordered ferroelectric state. In the third part, we consider the dynamic Jahn–Teller effect of Mn^3+^ cations in octahedral coordination at intermediate temperatures *T*_2_ < *T* < *T*_1_. The fourth part presents an analysis of the temperature dependence of the spontaneous polarization in BiMn_6.94_Fe_0.04_O_12_ at temperatures *T*_N1_ < *T* < *T*_3_ and *T*_3_ < *T* < *T*_2_. The final part describes the hyperfine magnetic interactions of ^57^Fe probes in the magnetically ordered temperature range *T* < *T*_N1_.

## 2. Results and Discussion 

### 2.1. Crystallographic, Magnetic, and Thermodynamic Data

X-ray diffraction patterns show no additional reflections corresponding to impurity phases (Appendix A). Having been measured at different temperatures, they suggest that the studied sample retains all the crystal modifications characteristic of an undoped (without Fe) quadruple manganite BiMn_7_O_12_ [3]. The observed reflections at 615 K are associated with the cubic (Im3¯) BiMn_7_O_12_ phase that is stable at *T* > *T*_1_ (Figure 1a). A part of the reflections split upon transition below *T*_1_, which corresponds to the monoclinic symmetry *I*2/*m* (Figure 1b). The monoclinic phase reflection (242) splits upon the further cooling of the sample (Figure 1c). As noted in [3], the temperature *T*_3_ of the phase transition monoclinic (*Im*) ↔ triclinic (*P*_1_) (Figure 1d) was not detected in the thermodynamic measurements; however, it can be evaluated from the deviation of the α and γ monoclinic unit cell angles from 90° (Appendix A). The estimated point *T*_3_ ≈ 240 K for BiMn_6.96_Fe_0.04_O_12_ is noticeably lower than ~290 K for the undoped BiMn_7_O_12_ sample [3]. 

The BiMn_6.96_Fe_0.04_O_12_ powder is characterized by a high degree of crystallinity according to the SEM data (Appendix A). It is inferred from particle agglomeration and the wide distribution of particle sizes ranging from 0.5 to 20 μm. Almost all crystallites have an irregular shape.

The peaks observed in the differential scanning calorimetry (DSC) curves of BiMn_6.96_Fe_0.04_O_12_ (Figure 2a) correspond to the phase transitions at the temperatures *T*_1_ ≈ 580–590 K and *T*_2_ ≈ 420–440 K, i.e., structural transitions *I*2/*m* ↔ Im3¯ and *Im* ↔ *I*2/*m*, respectively, according to the literature data [3]. It is worth noting that the undoped sample BiMn_7_O_12_ demonstrated the same transitions at ~ 608 K and ~ 460 K, respectively, as was reported in the earlier experiments [3]. Further lowering the lattice symmetry to *P*1 does not affect DSC curves. Measurements upon cooling and heating reveal a difference in the transition points Δ*T*_1_ ~ 7 K and Δ*T*_2_ ~ 20 K, both of which slightly exceed the corresponding values for the BiMn_7_O_12_ sample [3].

By measuring the heat capacity *C*_P_/T(*T*) in BiMn_6.96_Fe_0.04_O_12_ (Figure 2b), we obtained the values of the temperatures of transition to the magnetically ordered states *T*_N2_ and *T*_N3_. *T*_N2_ reached ~ 50 K, and the temperature of the third magnetic transition *T*_N3_ ≈ 24 K was 3–5 K lower than that of the undoped manganite [4]. The third phase transition at *T*_N3_ is also clearly seen in the temperature profile of the magnetic susceptibility *χ*(*T*) and is typical of antiferromagnets (Figure 2c). The parameters of the Curie–Weiss fit (Figure 2c) are in good agreement with the data obtained for the undoped manganite BiMn_7_O_12_ [3]. The temperature shift of structural and magnetic transitions upon iron-doping can be caused by the stabilization of a small quantity of iron in the manganite structure, rather than the precipitation of an impurity iron phase or its localization on the crystallite surface.

### 2.2. Mössbauer Data for T > T_1_

Figure 3a represents the typical Mössbauer spectra of ^57^Fe in BiMn_6.96_Fe_0.04_O_12_ measured at high temperatures *T* > *T*_1_. At these temperatures, the spectra consist of a quadrupole doublet with small and virtually temperature-independent splitting Δ ≈ 0.26 mm/s (Figure 4). The value of the isomer shift *δ*_633K_ ≈ 0.16 mm/s corresponds to Fe^3+^ cations [38], isovalently substituting Mn^3+^ in the octahedral positions of Mn2 (Figure 1a). Despite BiMn_6.96_Fe_0.04_O_12_ having a cubic structure (Im3¯) at *T* > *T*_1_ and the local octahedral anion environment of the Mn2 sites being formally considered to be undistorted, the local symmetry of the oxygen sites explains the non-zero quadrupole splitting of the spectrum (Table 1). Although all the Mn2 sites substituted by Fe^3+^ probes are equivalent, the experimental spectra cannot be satisfactorily fitted using one doublet with unbroadened resonant lines. This suggests the presence of a distribution *p*(Δ) of Δ values (Figure 3a), i.e., that the crystalline environment of the ^57^Fe probes is not homogenous.

The EFG parameters were calculated within the “ionic model” to support this assumption. It takes into account monopole ($V_{ZZ}^{\mathrm{mon}}$) and dipole ($V_{ZZ}^{\mathrm{dip}}$) contributions from ions that are located in the non-centrosymmetric sites in BiMn_7_O_12_. Having stereochemically active 6*s*^2^ lone-pair electrons, Bi^3+^ cations are considered to mainly contribute to $V_{ZZ}^{\mathrm{dip}}$. The lone pair induces the displacement of Bi^3+^ cations from their centrosymmetric positions, which is equivalent to inducing the electric dipole moment *p*_Bi_. Therefore, only dipole contributions ($V_{ZZ,Bi}^{\mathrm{dip}}$) from Bi^3+^ were taken into account when calculating $V_{ZZ}^{\mathrm{dip}}$ using variable *p*_Bi_ values in further calculations. Additionally, the dipole moments ***p***_Bi_ were considered to be randomly oriented in a crystal lattice since BiMn_7_O_12_ is paraelectric at *T* > *T*_1_ [3]. See Appendix B for details.

The inclusion of the dipole contribution from Bi^3+^ allows us to achieve a good agreement between the theoretical and experimental values of the quadrupole splitting. The calculated dipole moment *p*_Bi_ ≈ 1.2 × 10^−29^ C·m lays within the range of the corresponding values *p*_Bi_ for other Bi^3+^ oxide compounds [10]. Most importantly, even with a random orientation of the *p*_Bi_ moments, the Mn2 sites become non-equivalent in terms of the induced lattice contribution $V_{ZZ,\mathrm{Bi}}^{\mathrm{dip}}$. This is, in essence, the main cause of the observed broadening of the spectra, i.e., the appearance of the *p*(Δ) distribution (Figure 3a). Using the calculated values of Δ^theor^ for each Mn2 site within the *P*1 pseudocell (see Appendix B, Appendix C and Appendix D), which is characterized by the peculiar relative orientation of the surrounding dipole moments *p*_Bi_, we calculated the mean value of the quadrupole splitting as well as the dispersion $D_{p}^{theor}$= 0.020 mm^2^/s^2^, which was found to be close to $D_{p}^{\exp}$ ≈ 0.017 mm^2^/s^2^ of the experimental (Table 1) distribution *p*(Δ).

Thus, the Mössbauer data indicate that, in the paraelectric cubic phase of BiMn_6.96_Fe_0.04_O_12_ at *T* > *T*_1_, Bi^3+^ cations exist in a locally distorted environment and retain their electric dipole moments *p*_Bi_ that are randomly oriented in the cubic lattice. In this case, transitions to the anti- or ferroelectric states at lower temperatures *T* < *T*_2_ should be accompanied by the ordering of the *p*_Bi_ dipoles, i.e., they should represent the “order-disorder” phase transitions [39], as an alternative to the “displacive” phase transitions [40]. Previously, in refs. [41,42], static dipole moments *p*_Bi_ are assigned to the lone *sp^x^*-hybrid electron pairs of Bi^3+^ cations, which are oriented parallel to the direction of the displacement of a bismuth cation from its conventional centrosymmetric site (Figure 5a). Such an approach can qualitatively explain the unusually large thermal ellipsoids of Bi^3+^ cations reported in [1,3] for BiMn_7_O_12_ at *T* > *T*_1_. These ellipsoids may form as a result of the superposition of multidirectional *sp^x^*-hybrid pairs, whose randomly oriented lobes create a sphere that manifests itself in the diffraction patterns as unusually large bismuth thermal ellipsoids (Figure 5b). However, it should be noted that this approach is a simplified, albeit illustrative, model that has not been experimentally confirmed for the majority of known Bi(III) phases [43,44,45]. 

### 2.3. Mössbauer Data for T_2_ < T < T_1_

BiMn_6.96_^57^Fe_0.04_O_12_ undergoes the structural transition at *T*_1_ ≈ 590 K, transforming from cubic ($Im\bar{3}$) to monoclinic (*I*2/*m*) lattice symmetry (Figure 1b) with decreasing temperature. Figure 3b illustrates a typical Mössbauer spectrum of ^57^Fe probes in the monoclinic BiMn_7_O_12_, which has the shape of a symmetrically broadened quadrupole doublet. Despite the lowering of the manganite lattice symmetry, the obtained distributions *p*(Δ) show a single maximum that corresponds to the average <Δ> value which increases drastically upon decreasing temperature (Figure 3 and Figure 4). Considering the fact that the main contribution to the EFG imposed on the spherical Fe^3+^ cations is accounted for by the distortion of their crystalline surrounding (lattice contribution), it is difficult to explain the observed sharp change in the quadrupole splitting with temperature.

The results of the calculation of the EFG parameters for the different sites of Mn^3+^ with the monopole contributions from all ions (Bi^3+^, Mn^3+^, and O^2−^), as well as additional dipole contributions from Bi^3+^ and O^2−^, show that the values *V*_ZZ,Mn4_ = 3.76 × 10^20^ V/m^2^ and *V*_ZZ,Mn5_ = 4.21 × 10^20^ V/m^2^ are close to each other, which is probably responsible for the presence of only one maximum in *p*(Δ) (Figure 3b). As expected, the EFG parameters for both Mn sites are almost independent of temperature. Moreover, it was established that the calculated values Δ^theor^_Mni_ ≈ *eQ*$V_{ZZ,\mathrm{Mni}}^{Teop}$ for sites Mn4 and Mn5 (where *Q* is the quadrupole moment of ^57^Fe nuclei) remarkably exceed the corresponding experimental values Δ^exp^ (Figure 4; Appendix A).

We suppose that the abovementioned discrepancy between the calculated and experimental values of the quadrupole splitting (Δ^theor^ > Δ^exp^) and their unusually strong temperature dependences can be attributed to the dynamic JT effect of Mn^3+^ cations occurring in this temperature range [46]. The JT interactions of the Mn^3+^ cations in BiMn_7_O_12_ can result in the so-called orbital ordering, or cooperative JT effect, which is also observed in other perovskite-like Mn(III) oxide systems, namely, *R*MnO_3_ [47,48], *R*_1−x_*A*_x_MnO_3_ [11], and *A*Mn_7_O_12_ [37,49] (*R* = REE, *A* = Ca, Sr, Pb). All these systems can exhibit a structural transition to a crystal lattice with enhanced symmetry in the temperature range *T* > *T*_JT_, which is ascribed to the dynamic JT effect, or the “melting” of the cooperative JT distortion [46]. Similar phase transitions can occur through two mechanisms: one involves increasing the symmetry of distorted (Mn^3+^O_6_) polyhedra until the uniform population of Mn^3+^ *e_g_*-orbitals is achieved, and the other entails the orientational disordering of distorted (Mn^3+^O_6_) polyhedra while preserving the polarization of *e*_g_-orbitals even at high temperatures (*T* >> *T*_JT_) [50]. As was noted in several refs. [51,52,53], the local disordering of (Mn^3+^O_6_) polyhedra can start at a temperature (*T**) that is significantly lower than the temperature of the structural phase transition *T*_JT_ (>>*T**). However, there is still no reliable experimental data on the changes in the structure and electronic state of manganite, which take place in this “intermediate” temperature range.

Local minima are known to appear on the adiabatic potential surface of possible nuclear configurations of O^2−^ anions in the (MnO_6_) polyhedra if the anharmonicity of vibronic interactions is taken into account. These minima reflect the specific orthorhombic distortions of the corresponding polyhedra. With increasing temperature, the crystalline environment of the JT Mn^3+^cations stochastically relaxes between these minima due to thermally activated excitations or the tunnel effect [54]. Although the Fe^3+^ cations per se do not participate in the vibronic interactions, their local crystal environment also fluctuates dynamically due to the cooperative JT effect. Therefore, we suggest that the observed significant reduction in the Δ^exp^ values compared to the theoretical calculations can be associated with the relaxation behavior of the Mössbauer spectra in the temperature range *T*_2_ < *T* < *T*_1_. In [54,55], it was shown that such spectra can be described with the “two-level model” in the limit of “fast relaxation”, i.e., when Ω_R_ >> Ω_0_, where Ω_R_ and Ω_0_ are the frequencies of the relaxation of the oxygen environment and the precession of the ^57^Fe quadrupole moment around the *V*_ZZ_ direction, respectively. The model adopts the frequencies of forward (Ω_12_) and reverse (Ω_21_) transitions between states “1” and “2” as variables connected by the detailed equilibrium principle *n*_1_Ω_12_ = *n*_2_Ω_21_, where *n*_1_ and *n*_2_ are the populations of states (Figure 6a) [55].

In the monoclinic structure (*I*2/*m*) of BiMn_7_O_12_, the distortion of (MnO_6_) polyhedra corresponding to the energy minimum *E*_1_ of the adiabatic potential, is described as a “bonding” *Q*^(−)^—the linear combination of the orthorhombic (*Q*_2_) and tetragonal (*Q*_3_) vibrational modes [7,56]. In this case, the distortion with a higher energy *E*_2_ is attributed to the “antibonding” vibration mode *Q*^(+)^. In a local approximation, when only the closest anion environment is considered, the two vibrational modes—*Q*^(−)^ and *Q*^(+)^—correspond to the distortions that exhibit equal magnitudes but opposite signs in their EFG components (*V*_ZZ_) imposed on the ^57^Fe nuclei occupying the Mn4 and Mn5 sites [54]. Consequently, when the population of the *E*_1_ and *E*_2_ levels equalizes with increasing temperature, quadrupole splitting sharply decreases, i.e., Δ(*T*) ∝ <*V*_ZZ_>, where <*V*_ZZ_> is averaged over the energy states *E*_1_ and *E*_2_ [54]. On the other hand, the monotonous decrease in Δ(*T*) up to *T*_1_ can suggest a gradual enhancement of the (FeO_6_) symmetry upon approaching the temperature of the structural phase transition *I*2/*m* → Im3¯. This conclusion is consistent with the synchrotron X-ray diffraction studies of BiMn_7_O_12_, which also demonstrate the gradual decrease in the distortion parameter Δ_d_ of (MnO_6_) polyhedra when *T* → *T*_1_ [3]. Thus, these parameters behave similarly when assessed by inherently different characterization techniques. These data suggest the second-order JT phase transition that can be referred to as the displacive structural transition, in contrast to the “order-disorder” transition mechanism.

The fitting of the whole series of spectra within the framework of a two-level model allowed us to estimate the average relaxation frequency Ω_R_ = Ω_12_Ω_21_/(Ω_12_ + Ω_21_) ≈ (2–7) × 10^7^ Hz, which significantly exceeds the frequency of the quadrupole precession Ω_0_ ≈ 8.5 × 10^6^ Hz. Increasing temperature leads to a gradual equalization of the populations *n*_1_ and *n*_2_. This should result in a sharp decrease in the quadrupole splitting and a slight broadening of the doublet components in the limit of fast relaxation Ω_R_ >> Ω_0_. Indeed, this pattern describes the temperature-related changes of all the spectra at *T*_2_ < *T* < *T*_1_ (Appendix A). Using the linear approximation of ln(*n*_1_/*n*_2_) = f(1/*T*) (obtained from the Arrhenius equation): $$\Omega_{12(21)}(T)=\Omega_{1(2)}^{0}\exp\left(-\frac{E_{12(21)}}{k_{B}T}\right),$$
where $\Omega_{1(2)}^{0}$ are temperature-independent parameters; *E*_12(21)_ are the activation energies of “1” and “2” states, respectively; *k*_B_ is the Boltzmann constant) we evaluated the energy difference between the relaxed states Δ*E* = 69(2) meV and the mean value of the activation energy *E*_act_ = 220(9) meV (Figure 6a and Appendix A) that closely corresponds to <*E*_act_> for other perovskite-like Mn(III) manganites [57]. However, a deviation from linearity is observed at higher temperatures (*T* > 550 K) (Figure 6b). This is likely to result from the changes in the relative position of levels *E*_1_ and *E*_2_ between which the relaxation occurs. This explanation is indirectly supported by a similar temperature profile of the distortion parameters Δ_d_ of polyhedra (MnO_6_) (Figure 6b), which govern the splitting of the 3*d* levels of Mn cations under the influence of the ligand field.

It is worth noting that the observed structural changes in BiMn_7_O_12_ in the temperature range of the JT transition are similar to those in the isostructural phase of LaMn_7_O_12_ [58] but differ crucially from the so-called “traditional” perovskites *R*MnO_3_ (*R* = REE), in which the cooperative JT effect follows the “order-disorder” mechanism [39]. In these oxides, polyhedra (Mn^3+^O_6_) remain distorted even at temperatures significantly exceeding *T*_JT_. However, these distortions are randomly oriented in the crystal lattice, thus making the structure “macroscopically” more symmetrical compared to the low-temperature phase with orbital ordering.

### 2.4. Mössbauer Data for Temperature Ranges T_3_ < T < T_2_ and T_N1_ < T < T_3_

In the temperature range *T*_3_ < *T* < *T*_2_, the Mössbauer spectra of BiMn_6.96_Fe_0.04_O_12_ consist of a broadened quadrupole doublet (Figure 3c). The observed bimodal profile of *p*(Δ) (Figure 3c) suggests the stabilization of ^57^Fe nuclei at two different crystallographic sites of the manganite. This conclusion agrees with the earlier structural data for BiMn_7_O_12_ [3]: in the crystal lattice with the *Im* symmetry, the JT Mn^3+^ cations occupy two sites, Mn1 and Mn2, in the very distorted octahedral oxygen environment (Figure 1c). Thus, the high value <Δ> ≈ 0.62 mm/s observed in the spectra of Fe^3+^ probe cations is conditioned by a low symmetry of the oxygen environment of the JT Mn^3+^ cations. Based on the *p*(Δ) distribution, we fitted the whole series of spectra measured in the range *T*_3_ < *T* < *T*_2_ as a superposition of two quadrupole doublets, Fe(1) and Fe(2), having close values of isomer shifts, *δ*_1_ ≈ *δ*_2_ (Figure 3c).

The analysis above yielded anomalously sharp temperature dependences Δ_i_(*T*) for both Fe(1) and Fe(2) doublets (Figure 4). Such behavior can stem from the induction of spontaneous BiMn_7_O_12_ polarization at *T* < *T*_2_ [3]. To support this assumption quantitatively, we obtained an expression relating the lattice contributions $V_{ZZ}^{\mathrm{lat}}$ with the values and mutual orientation of electric moments (***p***_k_) in the lattice of BiMn_7_O_12_ (for details, see *SI*). The values ***p***_k_ and their projections *p*_ik_ were calculated using the Born model [59]: ***p***_k_
*= Z*_k_Δ***r***_k_ or *p*_ik_
*= Z*_k_Δ*x*_ik_, where *Z*_k_ is the Born charge of the *k*th ion, which is an isotropic scalar value in our calculation; Δ***r***_k_ (Δ*x*_ik_) is the vector of the displacement of the *k*th ion (and *i*th projection) from the centrosymmetric position. The values Δ***r***_kk_ and Δ*x*_ik_ were calculated using the crystallographic data for BiMn_7_O_12_ obtained at 300 K [3]. To estimate Born charges, we sequentially varied the charges {*Z*_Bi_, *Z*_Mn(i)_, *Z*_O(i)_} and their corresponding dipole moments *p*_ik_ with the given displacement Δ*x*_ik_ until the best agreement with the experimental splitting Δ_i_ was achieved (Appendix A). The values approximated in such a manner that *Z*_Bi_ = +3.30, *Z*_Mn1_ ≈ *Z*_Mn2_ = +3.30, and *Z*_O_ = −2.20, and all lie within the range of the Born charges obtained earlier for the corresponding ions in other perovskite oxides [60].

Using the above approximations, we derived an equation that describes Δ(*T*) as a function of the *i*th projections ${\left\{P_{i}\right\}}_{i=x,y,z}=\sum_{k}p_{ik}$ of the spontaneous polarization *P*_s_ = (*P*_x_^2^ + *P*_y_^2^ + *P*_z_^2^)^1/2^: (1)$$\Delta (T)=(1-\gamma_{\infty })\frac{\mathrm{eQ}}{2}\sum_{k}\frac{\partial^{2}}{\partial x_{i}^{2}}\left[\frac{q_{k}}{r_{k}}+\frac{r_{ik}P_{i}(T)}{r_{k}^{3}}\left(\frac{p_{k}(T_{0})}{P_{s}(T_{0})}\right)\right]$$
where *p*_k_(*T*_0_)/*P*_s_(*T*_0_) is the ratio of the dipole moment of the *k*th ion (*p*_k_) to the spontaneous polarization in the crystal, which is calculated based on the crystallographic data for BiMn_7_O_12_ (*Im*). The first and the second terms in Equation (1) are the monopole and dipole contributions to the EFG. Using Equation (1), we plotted theoretical dependencies Δ_1_(*P*_s_) and Δ_2_(*P*_s_) (Figure 7). Using the numerical solution of the equations Δ_1_(*P_s_*) = Δ_1_(*T*) and Δ_2_(*P_s_*) = Δ_2_(*T*) and accounting for statistical errors allowed us to simulate the temperature dependence *P*_s_(*T*) (Figure 8).

The same algorithm for constructing dependencies *P*_s_(*T*) using experimental data Δ_i_(*T*) was applied to the triclinic phase (*P*1) of BiMn_7_O_12_. The distribution *p*(Δ) has a trimodal profile for the given structure modification (Figure 3d), which indicates that Fe^3+^ cations occupy at least three nonequivalent sites. According to the structural data [4], the Mn^3+^ cations form four equally populated sites (Mn4, Mn5, Mn6, and Mn7) in an octahedral oxygen environment (Figure 1d). The calculated EFG parameters of Mn4 and Mn6 suggest that these atoms are located symmetrically near similar crystalline environments, making them indistinguishable in the ^57^Fe Mossbauer spectra. Therefore, the spectra at *T*_N1_ ≤ *T* ≤ *T*_3_ were fitted with three quadrupole doublets, namely, Fe(1), Fe(2), and Fe(3), with the Fe(3) component being two times more intense than Fe(1) and Fe(2) (Figure 3d). Using the structural data for the triclinic BiMn_7_O_12_ phase at 10 K [4] and the described algorithm combining the theoretical dependencies Δ_i_(*P*_s_) with the experimental Δ_i_(*T*) values, we were able to model *P*_s_(*T*) across the temperature range under investigation (Figure 8).

The dependences Δ ∝ *V*_ZZ_ = f(*P*_s_) (Figure 7) agree with the results of [24,25,26,27], where, in a general case, the dependence of *V*_ZZ_(*P*_s_) at low *P*_s_ values was represented using a Taylor series expansion:(2)$$V_{\mathrm{zz}}(P_{s})=V_{\mathrm{zz}}^{\left(0\right)}+\sum_{i}\left(\frac{\partial V_{\mathrm{zz}}}{\partial x_{i}}\right)P_{s}+\frac{1}{2}\sum_{i,j}\left(\frac{\partial^{2}V_{\mathrm{zz}}}{\partial x_{i}\partial x_{j}}\right)P_{s}^{2}+\ldots =V_{\mathrm{zz}}^{\left(0\right)}+\;\alpha P_{s}+\;\;\beta P_{s}^{2}+\;\gamma P_{s}^{4}\;\ldots$$

It was shown in [12] that the linear term vanishes and the quadratic term in the expansion becomes significant (*β* >> *γ* ≠ 0) for the centrosymmetrical crystal sites. At the same time, if the center of symmetry is absent, the linear term should be predominant (*α* >> *β* >> *γ* ≠ 0). Since for both polymorphic modifications of the BiMn_7_O_12_ octahedral Mn^3+^ sites are not centrosymmetric, the above dependences *V*_ZZ_(*P*_s_) can be described using an expansion in a series with nonzero parameters *α* and *β*, whose values (Table 2), in order of magnitude, are consistent with similar data from other perovskite-like ferroelectrics [24,25,26,27].

When describing the dependences, *P*_s_(*T*) was obtained for two temperature ranges, and we used the model of the average effective field [61]. Within the framework of this approach, it is assumed that every ion in the ferroelectric crystal is affected by an effective electric field (*E*_eff_), which can be expressed as
(3)$$E_{\mathrm{eff}}=E_{0}\;+\beta P_{s}+\;\gamma P_{s}^{3}+\;\delta P_{s}^{5}+\ldots$$
where *E*_0_ is the external electrical field, and the following terms correspond to the dipolar, quadrupolar, and octupolar interactions, respectively. In our calculations, *E*_0_ = 0. Moreover, only dipolar and quadrupolar interactions were taken into consideration, the latter (*γ*) serving to describe the phase transitions of both the first and second orders within the united approach. For statistical consideration, the *P*_s_(*T*) dependence of polarization can have a general form [61]:(4)$$P_{s}(T)=P_{0}\tanh\left(\frac{E_{\mathrm{eff}}P_{0}N^{-1}}{k_{B}T}\right)=P_{0}\tanh\left(\frac{(\beta P_{s}+\gamma P_{s}^{3})P_{0}N^{-1}}{k_{B}T}\right)$$
where *P*_0_ is the spontaneous saturation polarization, and *N* is the number of elementary dipoles per unit cell. Using the relationship *k*_B_*T*_C_ = *β*·*P*_0_^2^/*N* (where *T*_C_ is the Curie ferroelectric point) and designations of the normalized values *σ*_s_ ≡ *P*_s_/*P*_0_, *τ* ≡ *T*/*T*_C_, *g* ≡ *γP*_0_^2^/*β*, one can derive an expression that is convenient for analyzing the experimental data:(5)$$\sigma_{s}(T)=\tanh\left(\frac{\sigma_{s}(T)\;+\;g\sigma_{s}^{3}(T)}{\tau }\right)$$
where the parameter *g* is a quantitative criterion of the order of the transition to the ferroelectric state [61]. The analysis of the experimental dependences *P*_s_(*T*) using Expression (5) is shown in Figure 8.

The value *g* ≈ 0.48(3) in the temperature range *T*_3_ < *T* < *T*_2_ indicates a significant contribution of the quadrupolar interactions (*γ* ≠ 0) in Expression (3), which, in turn, is a feature of the first-order transitions [61,62]. A similar behavior was observed for many oxide systems, in particular, those demonstrating multiferroic properties [63]. The transition point *T*_C_ ≈ 437 K of the ferroelectric state, having been evaluated within the framework of this approach based on the description of *P*_s_(*T*), insignificantly differs from the temperature of the structural transition *Im* ↔ *I*2/*m* (*T*_2_ ≈ 442 K) determined for the sample BiMn_6.96_Fe_0.04_O_12_ from the thermodynamic data (Figure 2a). We suggest that the observed first-order transition at *T*_2_ stems from an inevitable coupling between the electric polarization and the crystal lattice. There is a simultaneous structural change accompanying this transition as evidenced by the above finding. By using a simple phenomenological approach (see Appendix E), it can be shown that the coupling between spontaneous polarization (*P*_s_) and strain (*ε*) can switch an otherwise second-order transition to a first-order transition. Moreover, there is a relationship between the strength of the ferroelastic coupling and the size of the hysteresis ~20 K (Figure 2a) of the resultant first-order transition. It is known that the size of hysteresis is determined by the energy barrier at *T*_C_ (Appendix A), which is largely dependent on the magnitude of the ferroelastic coupling coefficient λ in the fourth-order term of the Landau free energy (see Equation A8). On the other hand, a larger λ also leads to a larger spontaneous lattice distortion upon the first-order phase transition. Therefore, the magnitude of thermal hysteresis increases with an increase in lattice distortion.

In the range *T*_N1_ ≤ *T* ≤ *T*_3_, the dependence *P*_s_(*T*) demonstrates a kink at the point *T*_3_, and its course follows Expression (4) with the parameter *g* ≈ 0, which, upon extrapolation to ~294 K (see Figure 8), should correspond to a “gradual” second-order phase transition [61]. A discussion of the nontrivial course of the dependencies *P*_s_(*T*) is beyond the scope of our work; however, it may motivate someone to study this unusual system using new independent local and macroscopic methods and theoretical approaches.

### 2.5. Mössbauer Study in the Temperature Range T < T_N1_

The quadrupole doublets at temperatures slightly below the Néel point (*T* < *T*_N1_ ≈ 50 K) contain the broadened components that reflect hyperfine magnetic fields *B*_hf_ induced on the ^57^Fe nuclei (Figure 9a). The spectra were fitted via a reconstruction of distributions *p*(*B*_hf_) characterized by a certain dispersion *D_P_*(*δ*) at a given temperature. The kink on the temperature dependence *D_P_*(*δ*) = f(*T*) (Figure 9b; Appendix A for data) corresponds to the temperature 57(3) K, which, within the measurement error, coincides with *T*_N1_ ≈ 59 K for undoped manganite BiMn_7_O_12_ (Figure 2b). This result gives independent confirmation of the stabilization of ^57^Fe probes in the lattice of the BiMn_7_O_12_ manganite under study.

At low temperatures, *T* << *T*_N1,_ the Mössbauer spectrum of BiMn_6.96_^57^Fe_0.04_O_12_ has an asymmetric and slightly broadened Zeeman structure (Figure 10), which can be described by a superposition of four unequally broadened sextets in accordance with the structural data for the triclinic phase BiMn_7_O_12_ [4]. With increasing temperature, the profiles of these sextets change noticeably, which is characteristic of the system exhibiting relaxation processes. Earlier studies showed [64] that this behavior could be attributed to the magnetic excitation of paramagnetic impurity ions within magnetically ordered matrices, where competing exchange interactions play a significant role. When embedded within the manganite matrix, impurity cations Fe^3+^ with half-filled orbitals are surrounded by the JT Mn^3+^ cations with anisotropic orbital occupation. This can lead to a noticeable weakening of the exchange interactions between the impurity cations and their magnetic environment. Essentially, this suggests that iron cations can undergo lower-energy magnetic excitations, influencing also the neighboring Mn^3+^ cations rather than only Fe^3+^ cations. An increase in temperature leads to the progressive occupation of magnetic Fe^3+^ states characterized by different projections (*S*_Z_) of the total spin *S* = 5/2. As the magnetic interaction of iron with its surroundings is weakened, the relaxation of spin between the states |5/2, *S*_Z_> decelerates. If the relaxation period *τ*_R_ closely corresponds to the period of Larmor precession (*τ*_L_) of the ^57^Fe nuclear spin around the hyperfine field *B*_hf_, the Mössbauer spectra typically have complex relaxation profiles [65].

Additionally, it should be noted that the SEM data indicates that an average particle size exceeded ~2 µm for the BiMn_6.96_Fe_0.04_O_12_ sample (Appendix A). Consequently, the observed relaxation behavior of the Mössbauer spectra cannot be attributed to the superparamagnetic or superferromagnetic states of small particles [65,66].

Taking into account the considerations presented above, the spectra were fitted using a multilevel relaxation model [67]. This model is based on the assumption that in the effective magnetic Weiss field, spin *S* = 5/2 of the Fe^3+^ cation, stochastically relaxes between Zeeman states |5/2, *S*_Z_> [67]. Along with the static parameters (*δ*, *eQV*_ZZ_ and *B*_hf_), the model includes variable relaxation parameters, namely, the relaxation frequency Ω_R_ (=1/*τ*_R_) and the relative population (*s*) of the Zeeman sublevels between which the relaxation occurs. A detailed description of this model can be found in [67]. This allowed us to process the whole series of spectra measured in the temperature range 10 K < *T* < *T*_N1_. It should be noted that the static and relaxation parameters of the Mössbauer spectra remain virtually unchanged, which indicates the stability of a complex model of spectrum processing. A smooth and continuous change in the hyperfine magnetic field near *T*_N1_ indicates the occurrence of a second-order magnetic phase transition. This conclusion is consistent with the results of a theoretical study of undoped manganite BiMn_7_O_12_, according to which the transition at *T*_N1_ = 59 K corresponds to the formation of a single *E*-type AFM structure [68]. At the same time, the magnetic phase transitions at *T*_N2_ ≈ 50 K and *T*_N3_ ≈ 24 K cannot be seen in the Mössbauer spectra because the former does not change the local magnetic environment of manganese (probe iron) cations in *B* sites, and the second transition (*T*_N3_) leads to the magnetic ordering of Mn^3+^ in the *A*″ sublattice.

Characteristic of spin–spin relaxation, there are no obvious changes in the frequency Ω_R_ as a function of temperature. However, the values of Ω_R_ ~ (0.2–0.5) × 10^9^ s^−1^ are found to be significantly lower than the characteristic frequencies of spin waves Ω_S_ ≈ *k*_B_*T*/*ħ* = 10^11^–10^12^ s^−1^ (*T* = 1–100 K) in conventional magnetic systems [65]. This indicates that the spin fluctuations involving iron probes are predominantly local. Furthermore, the Ω_R_ value is comparable to the Larmor frequency Ω_R_ ≈ 10^8^ s^−1^ of the ^57^Fe nuclear spin in the hyperfine magnetic field <*B*_hf_> ≈ 50 T, thus supporting our assumption about the relaxed nature of the observed spectra.

Figure 11 shows the temperature dependence of the hyperfine field <*B*_hf_(*T*)> averaged over all partial spectra Fe(i), which was approximated for spin *S* = 5/2 using the parametric Brillouin function:(6)$$\langle B_{hf}(T)\rangle =\langle B_{hf}(0)\rangle B_{5/2}\left[\xi \frac{5}{2}\left(\frac{T_{N}}{T}\right)\sigma_{\mathrm{Mn}}(T)\right]$$
where *ξ* = *J*_FeMn_/*J*_MnMn_ is the ratio of exchange integrals that characterize the magnetic interactions of Fe^3+^ probes with the surrounding manganese cations (*J*_FeMn_) and the averaged interactions between the Mn^3+^ cations themselves (*J*_MnMn_). The value *ξ* = 0.67(3) obtained from the best fit of the experimental spectra evidences the weakening of the exchanged magnetic interactions of the iron cations with the manganese sublattice, which is equivalent to decreasing the effective Weiss field [69]. This can result from a so-called “orbital dilution”, a phenomenon characteristic of the impurity cations of transition metals with non-degenerate orbital electron states (Fe^3+^, Cr^3+^…: <*L*> = 0, where *L* is the total orbital momentum) if they are stabilized within the matrix of transition metals with degenerate orbital states (Rh^4+^, Mn^3+^…: <*L*> ≠ 0) [70,71]. It was shown previously that such impurity centers behave as peculiar “orbital defects” due to the absence of orbital degeneration, i.e., orbital degrees of freedom. Even at very low concentrations, they can significantly affect the magnetic structure of the compound. Experimental methods employed to study such systems are currently in their early stages of development.

## 3. Materials and Methods

The manganite BiMn_6.96_Fe_0.04_O_12_ was synthesized in a high-pressure, “belt”-type chamber. A stoichiometric mixture of Mn_2_O_3_ (99.9%, Rare Metallic Co., Tokyo, Japan) Bi_2_O_3_ (99.9999%, Rare Metallic Co., Tokyo, Japan), and ^57^Fe_2_O_3_ (95.5% enriched with ^57^Fe, Trace Sciences International, Richmond Hill, ON, Canada) was filled into a gold capsule in which it was subjected to a pressure of ~6 GPa followed by heating to 1323 K for 10 min. The sample was quenched to room temperature after holding for 120 min. The synthesis of undoped manganite BiMn_7_O_12_ is described in more detail in [3].

The X-ray diffraction data were acquired using a synchrotron source of X-rays (SXRPD) in a large Debye–Sherrer chamber with the line BL15XU (SPring-8, Sayo, Japan) and the 2*θ* value ranging from 3° to 60° with a step of 0.003°. The monochromatic radiation with the wavelength of *λ* = 0.65298 Å was used. Experiments were performed in a temperature range of between 100 and 670 K. Prior to measuring, the powder samples were tightly packed in a Lindemann glass capillary (for the 100–400 K range) and a quartz capillary (for the 350–670 K range) with an internal diameter of 0.1 mm. The capillaries were cooled using an N_2_ flow when the low-temperature measurements were performed. The processing of the diffraction patterns and refinement of the crystal lattice parameters were performed using the Rietveld method, using the RIETAN-2000 software similar to the procedure described in [3].

Scanning electron microscopy (SEM) images were taken on a NVision 40 electron microscope (Carl Zeiss; Oberkochen, Germany) equipped with an Oxford Instruments X-Max analyzer. The accelerating voltage varied in the range from 3 to 20 kV.

For measuring differential scanning calorimetry (DSC) curves on a Mettler Toledo DSC1 STAR^e^ calorimeter in the temperature range 125–673 K, samples were placed in Al crucibles, the rate of heating/cooling in the nitrogen flow being 10 K/min.

The heat capacity measurements were carried out on a PPMS calorimeter (Quantum Design, San Diego, CA, USA) in the temperature range of 2–300 K in the modes of heating and cooling in external magnetic fields ranging from 0 to 90 kOe.

Magnetic susceptibility was measured on a SQUID MPMS 1T magnetometer (Quantum Design, San Diego, CA, USA) in the temperature range of 2–350 K in the ZFC (cooling without external magnetic field) and FC (cooling in the external magnetic field 10 kOe in strength) modes.

Mössbauer spectra were measured with a conventional electrodynamic-type spectrometer in the constant acceleration mode with a 1450 MBq ^57^Co(Rh) γ-ray source. The values of the isomer shift are given relative to α-Fe (298 K). Processing of the experimental spectra was performed with the use of the program package “SpectrRelax” [72]. Computations of the EFG parameters were carried out using the “GradientNCMS” software (ver. 8.3) designed by the authors and are represented in more detail in [73].

## 4. Conclusions

We explored the interplay between the local crystal structure of the multiferroic BiMn_7_O_12_ manganite and the processes of its spontaneous polarization and magnetic ordering using ^57^Fe-probe Mössbauer spectroscopy. It was shown that Fe^3+^ probes statistically substitute isovalent Mn^3+^ cations in the octahedral oxygen local environment. The parameters of the electric hyperfine interactions of ^57^Fe nuclei reflect the symmetry of the crystalline environment of Mn^3+^ cations in these sites.

The calculations of the EFG parameters, considering both monopole and dipole contributions, are in accordance with our experimental results, demonstrating that, in the paraelectric phase (at *T* > *T*_2_), cations Bi^3+^, even while existing in locally distorted crystalline environments, maintain their electrical dipole moments *p*_Bi_, which are randomly oriented within the cubic lattice. As a result, the phase transitions into the ferroelectric state involve the ordering of *p*_Bi_ dipoles, i.e., they may be considered the transitions of the “order-disorder” type.

It was determined that the monotonous decrease in Δ(*T*) from *T*_2_ up to *T*_1_ can indicate a gradual increase in the symmetry of (Fe^3+^O_6_) polyhedra while approaching the temperature of the structural transition $Im\bar{3}$ → *I*2/*m*, which is corroborated by the synchrotron diffraction studies of undoped BiMn_7_O_12_. This characteristic behavior has been independently registered through methods that are entirely different in their physical nature. Thus, it strongly indicates the occurrence of the second-order JT phase transition. Its mechanism can be classified as a structural transition of the “displacive” type, in contrast to “order-disorder” transitions.

Using the Born model, we calculated dynamic ion charges that indicate only a minor ion polarization and the predominance of ionic contributions in the spontaneous electrical polarization of the crystal. The observed robust temperature dependence of the quadrupole splitting Δ_i_(*T*) of the partial spectra Fe(i) is governed by the temperature dependence *P*_s_(*T*). The dependence *P*_s_(*T*) on the opposite sides of the phase transition *P*1 ↔ *Im* (*T*_3_ ≈ 240 K) significantly differs in its behavior. In the range *T*_3_ < *T* < *T*_2_, *P*_s_(*T*) indicates that the ferroelectric–paraelectric phase transition is of the first order. The Curie point *T*_C_ ≈ 437 K determined from the Mössbauer data closely coincides with the temperature of the structural transition *Im* ↔ *I*2/*m.* The proposed algorithm for finding the correlation between the experimental dependencies Δ_i_(*T*) for the probe ^57^Fe nuclei and the polarization *P*_s_(*T*) of the crystal can be applied to other systems with ferroelectric and multiferroic properties.

At low temperatures, *T* < *T*_N1,_ and the ^57^Fe Mössbauer spectra demonstrate relaxation behavior. This can result from a so-called “orbital dilution” characteristic of the impurity cations of transition metals with non-degenerate orbital electron states within the matrix of transition metals with degenerate orbital states.

## Figures and Tables

**Figure 1 ijms-25-01437-f001:**
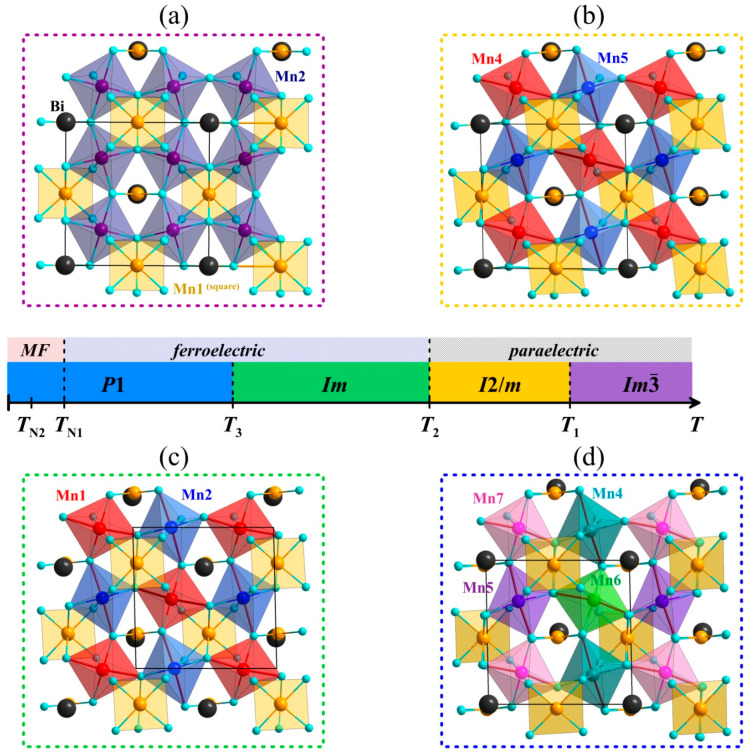
Crystal structures of BiMn_7_O_12_: (**a**) At *T* > *T*_1_ in the cubic $Im\bar{3}$ structure (without Bi splitting); (**b**) At *T*_2_ < *T* < *T*_1_ in the monoclinic *I*2/*m* structure; (**c**) At *T*_3_ < *T* < *T*_2_ in the monoclinic *Im* structure; (**d**) At *T* < *T*_3_ in the triclinic *P*1 structure (all viewed along the monoclinic *b* axis; Elongated Mn–O bonds due to the Jahn–Teller distortions in MnO_6_ octahedra are marked by red lines; The crystal cells are marked with black lined rectangles. The inset in the center depicts the accordance of the crystal structures and physical properties.

**Figure 2 ijms-25-01437-f002:**
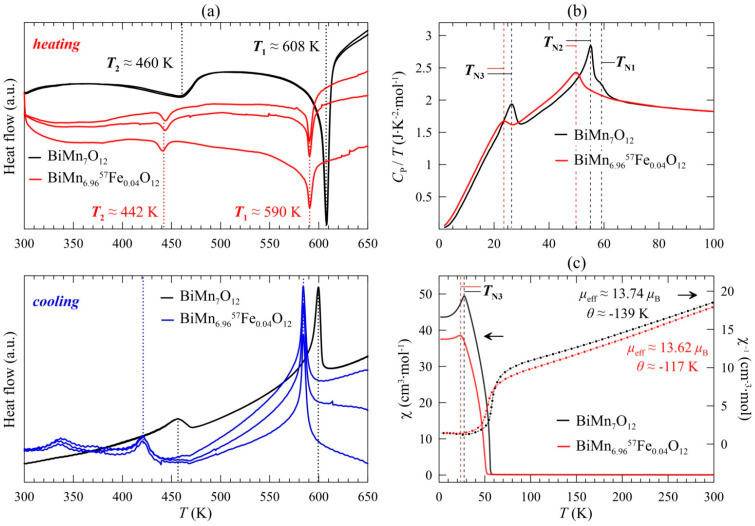
(**a**) Differential scanning calorimetry (DSC) curves of BiMn_6.96_^57^Fe_0.04_O_12_ upon heating and cooling (three runs were performed to check the reproducibility; since there were no peaks observed, data in the 125–300 K range are not shown); (**b**) Specific heat, plotted as *C*_P_/*T* versus *T*, of BiMn_7_O_12_ and BiMn_6.96_^57^Fe_0.04_O_12_ at *H* = 0 (measurements were performed on cooling); (**c**) ZFC dc (left scale) and reversed (right scale) magnetic susceptibility curves of BiMn_7_O_12_ and BiMn_6.96_^57^Fe_0.04_O_12_ (dashed vertical lines emphasize magnetic anomalies).

**Figure 3 ijms-25-01437-f003:**
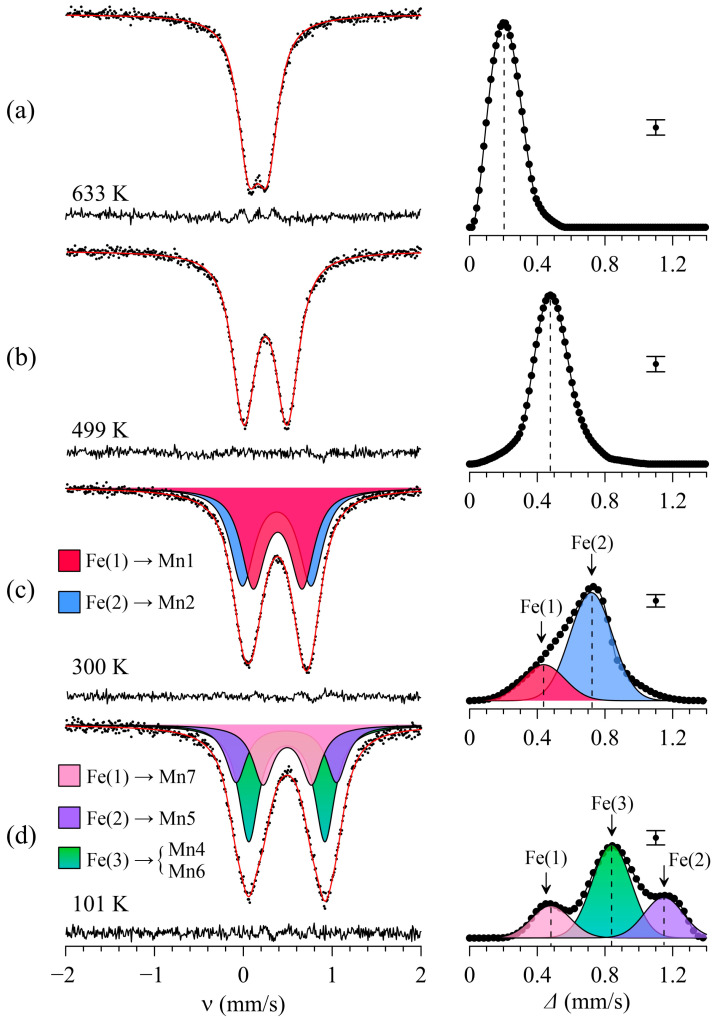
**Left panel**: typical Mössbauer spectra of the ^57^Fe nuclei in BiMn_6.96_Fe_0.04_O_12_ manganite measured at different temperatures (each at a specific range according to different crystal structures). **Right panel**: the *p*(Δ) distributions and their representation as the superposition of normal distributions corresponding to the crystal sites of ^57^Fe probe nuclei within a manganite structure. (**a**–**d**) Correspond to particular temperature range (see text).

**Figure 4 ijms-25-01437-f004:**
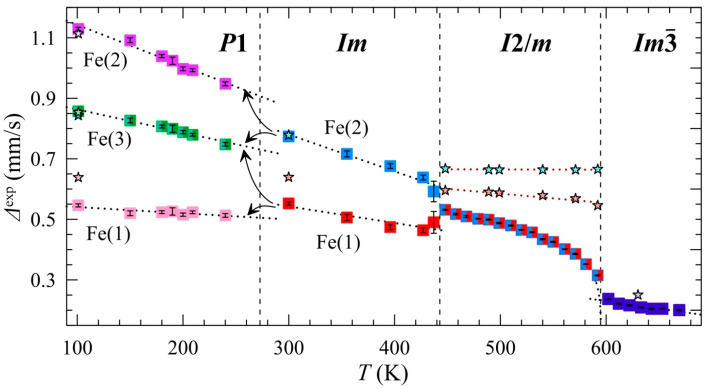
The temperature dependencies of the experimental values of the quadrupole splittings Δ_i_^exp^(*T*) of the partial Fe(i) spectra at specific ranges according to different crystal structures of the BiMn_6.96_Fe_0.04_O_12_ manganite (asterisks show the theoretical values).

**Figure 5 ijms-25-01437-f005:**
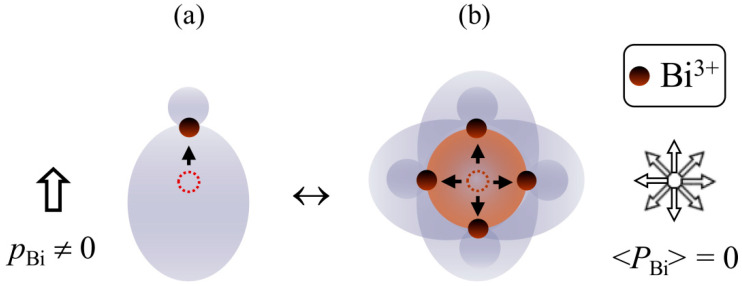
Schematic representations: (**a**) the formation of the *p*_Bi_ dipole moment as a result of the displacement of Bi^3+^ cations (brown balls) from their centrosymmetric positions (balls with a dotted line). The Bi^3+^ center is shifted toward the lone pair; (**b**) the random orientation of the lone electron pairs or displacements of Bi^3+^ cations leads to the zero value of the total crystal polarization (<*P*_Bi_>) averaged over all directions (the large brown ball represents the ellipse of the thermal vibrations of bismuth).

**Figure 6 ijms-25-01437-f006:**
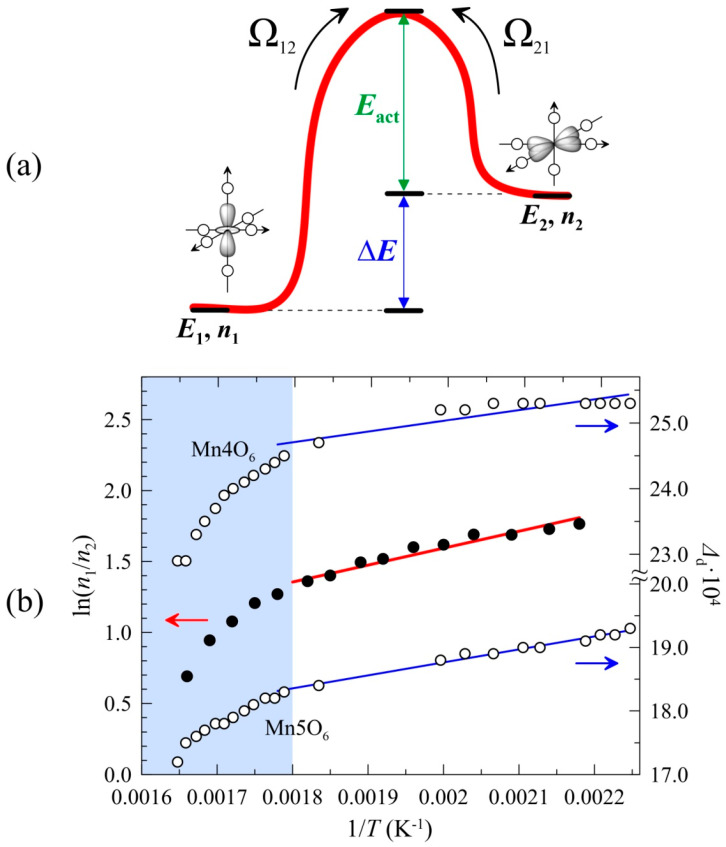
(**a**) The explicative scheme of the two-level relaxation model: *E*_i_—the energies of states “1” and “2”, *n*_i_—the probabilities of states, Ω_ij_—the frequencies of transitions between states. (**b**) The reciprocal temperature dependencies of the logarithm ln(*n*_1_/*n*_2_) of probabilities *n*_1_ and *n*_2_ ratio, and the distortion parameters Δ_d_ for Mn_4_O_6_ and Mn5O_6_ octahedra, calculated from structural data [3]. Blue lines represent the linear approximation in the selected temperature range and are shown for visual convenience. The shaded part corresponds to the temperature range (*T* > 500 K) for which a change in the degree of the distortion (Δ_d_) of the Mn*i*O_6_ polyhedra is expected and, as a consequence, so too is a change in the relative position of the energy levels *E*_1_ and *E*_2_ (see text) The black dots correspond to the left scale, and the circles correspond to the right scale.

**Figure 7 ijms-25-01437-f007:**
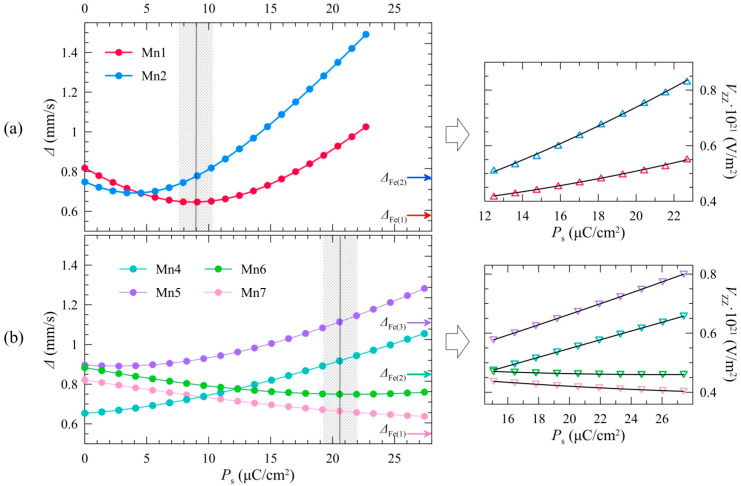
**Left panel**: The dependencies of the theoretical Δ_i_ values versus spontaneous crystal polarization *P*_s_ at (**a**) *T* = 300 K and (**b**) *T* = 10 K. The curves refer to the experimental values Δ^exp^_i_ from the Mössbauer spectroscopy data at (**a**) 300 K and (**b**) 101 K temperatures. Shaded areas correspond to evaluated *P*_s_ values when the theoretical Δ_i_(*P*_s_) values conform to the experimental Δ^exp^_i_ ones in the best way. The verticle lines in the shaded areas showed the approximate mean position for ease of perception. **Right panel**: the dependencies of the theoretical *V*_ZZi_ values versus spontaneous crystal polarization *P*_s_ at corresponding temperatures fitted with quadratic functions (see text).

**Figure 8 ijms-25-01437-f008:**
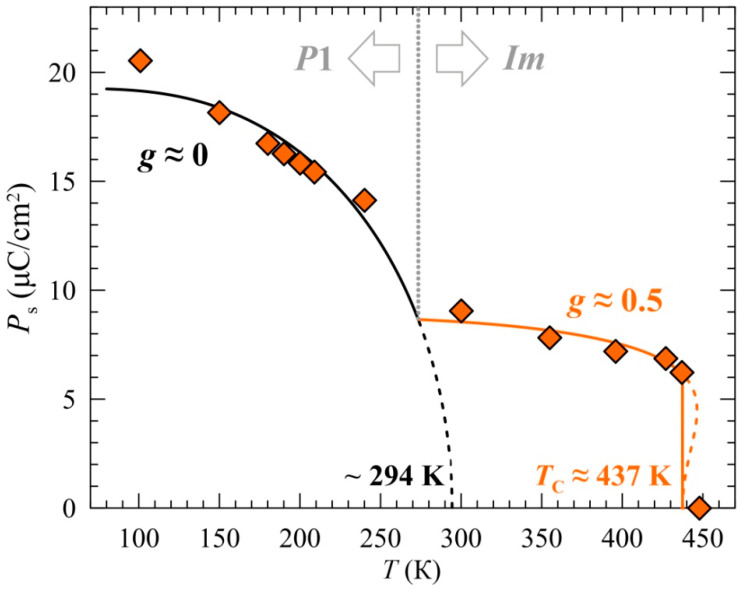
The temperature dependencies of the spontaneous polarization *P*_s_(*T*) for the two crystal structures of the BiMn_6.96_Fe_0.04_O_12_ manganite. The solid and dashed curves represent the fitting in order with theory explained in the text. The dotted line shows the temperature of the phase transition.

**Figure 9 ijms-25-01437-f009:**
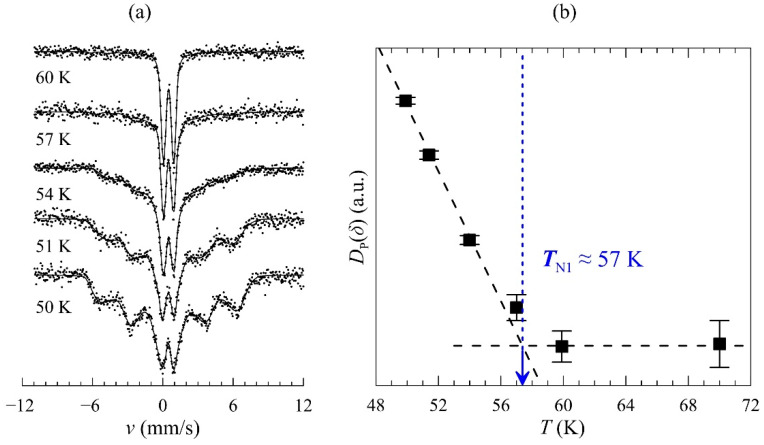
(**a**) ^57^Fe Mössbauer spectra of BiMn_6.96_Fe_0.04_O_12_ near *T*_N1_ fitted as the distributions of the single Lorentz line; (**b**) the temperature dependence of the dispersion *D*_P_(*δ*) of the isomer shift *δ*. The kink was used to evaluate the magnetic phase transition point (see text).

**Figure 10 ijms-25-01437-f010:**
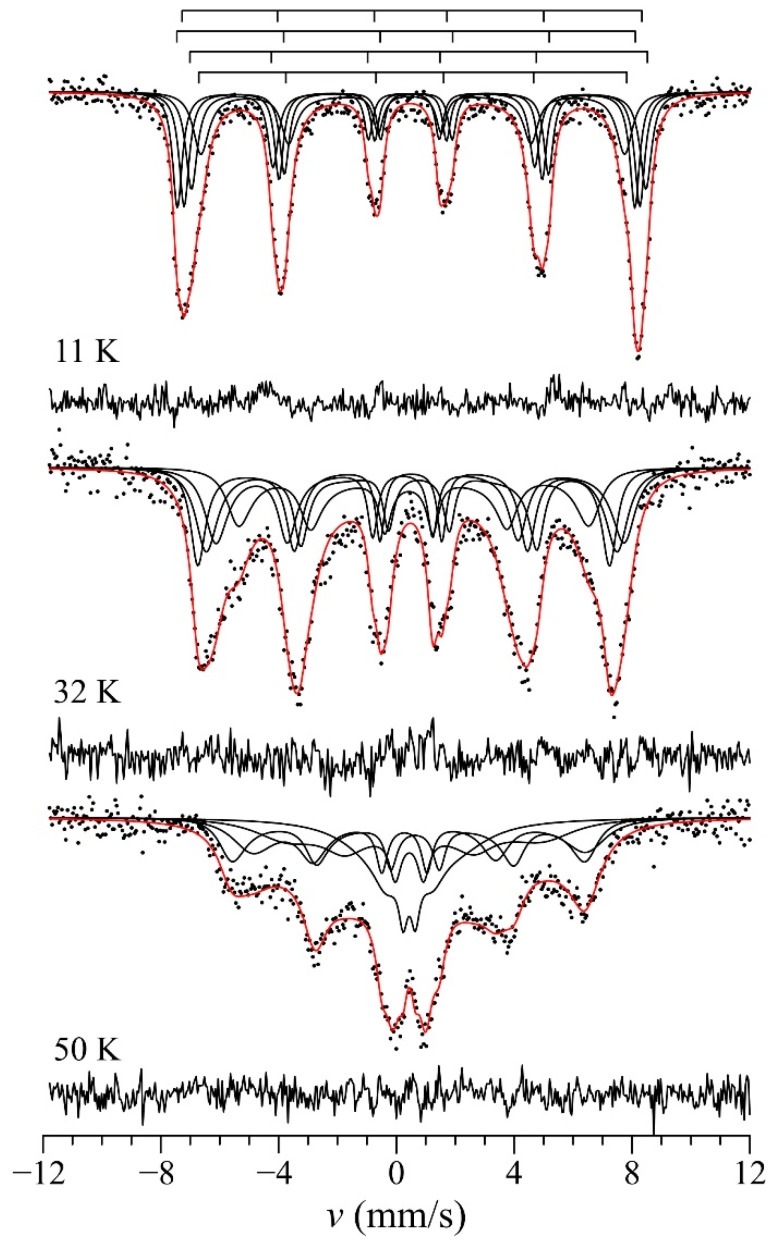
^57^Fe Mössbauer spectra of BiMn_6.96_Fe_0.04_O_12_ at *T* < *T*_N1_, fitted with the multilevel magnetic spin relaxation (see text). Black curves are subspectra obtained during fitting procedure, red curves are summarized fitted spectra. The differecnces between experimental and fitted data are also shown in the bottom of each spectrum.

**Figure 11 ijms-25-01437-f011:**
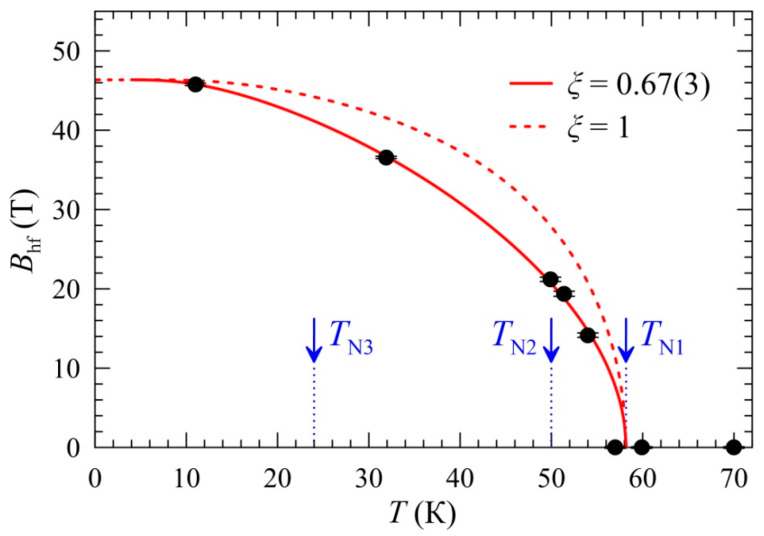
The temperature dependence *B*_hf_(*T*) of the hyperfine magnetic field *B*_hf_ at ^57^Fe nuclei approximated using a modified Brillouin function (see text). The dashed red line shows the “pure” Brillouin law for spin *S* = 5/2. Black dots are experimental mean values of the hyperfine fields.

**Table 1 ijms-25-01437-t001:** ^57^Fe hyperfine parameters at *T* > *T*_1_ of BiMn_6.96_^57^Fe_0.04_O_12_.

*T*, K	<*δ*>, mm/s	<Δ>, mm/s	$D_{p}^{\exp}$, mm^2^/s^2^	*Γ*, mm/s
602	0.18(1)	0.28(1)	0.016(1)	0.24 *
622	0.17(1)	0.26(1)	0.016(1)	0.24 *
633	0.16(1)	0.26(1)	0.017(1)	0.24 *
653	0.15(1)	0.25(1)	0.016(1)	0.24 *

* When processing the spectra, the linewidth *Γ* was fixed in accordance with the “thin” absorber and properties of the source. <*δ*> is the mean isomer shift, <Δ> is the mean quadrupole splitting, $D_{p}^{\exp}$ is the dispersion of the quadrupole splitting taken from the distribution reconstruction procedure.

**Table 2 ijms-25-01437-t002:** Taylor expansion parameters *V*_ZZ_^(0)^, *α*, and *β* of the dependences *V*_ZZ_(*P*_s_) calculated for both polymorphic modifications for all octahedral Mn^3+^ sites.

*T*, K	Site	*V*_ZZ_^(0)^ (V/m^2^ × 10^21^)	*α* (V/C × 10^21^)	*β* (V·m^2^/C^2^ × 10^22^)
300	Mn1	0.38(1)	−0.6(1)	0.78(3)
	Mn2	0.27(1)	1.9(1)	0.48(2)
101	Mn4	0.385(1)	0.71(1)	0.12(3)
	Mn5	0.519(3)	0.39(2)	0.24(6)
	Mn6	0.565(1)	−0.85(1)	0.195(1)
	Mn7	0.526(1)	−0.72(1)	0.100(1)

## Data Availability

The data presented in this study are available on request from the corresponding author.

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
