# Peer review of "Understanding Complex Interplay among Different Instabilities in Multiferroic BiMn7O12 Using 57Fe Probe Mössbauer Spectroscopy"

_ijms, 2024, doi:10.3390/ijms25031437_

Round 1

Reviewer 1 Report

Comments and Suggestions for Authors

This is a valuable work clearly written with sound coclusions.

I have some minor comments.

Please explain why the line width in Table 1 was fixed at 0.24 mm/s. (Why exactly 0.24?)

The residual spectrum  at 633 K (Fig. 3 a) shows substantial misfit at the peaks. This shows that even a QS distribution is not enough to fit the spectrum. Is it possible that the fixed 0.24 mm/s width is too large for this case?

When evaluating spectra with overlapped doublets, it is very important to make sure that there is no texture (preferred orientation of the crystallites of the Mössbauer sample). The Materials and Methods section is very short. The sample preparation should be discussed. The SEM image in the Suppl. shows granular particles (no needles or platelets), thus  the texture is unlikely, but it should be stated in a written form that the texture is excluded in the samples. Otherwise the 2-3 doublets seen in the spectra may be virtual.

This text: „The direction of the lone electron pair is shown, which is opposite to the direction of displacement of the Bi3+ cations” is not very clear to me. The lone pair has no direction itself, it has a position. I would simply say: the Bi3+ center moves/is shifted/is displaced toward the lone pair. (See lines 202, 212,213)

I recommend publication of this work after minor revisions.

Comments on the Quality of English Language

Minor language correction suggestions are indicated in the attached file.

Author Response

This is a valuable work clearly written with sound coclusions.

I have some minor comments.

Please explain why the line width in Table 1 was fixed at 0.24 mm/s. (Why exactly 0.24?)

Our response: This value corresponds to the linewidth for the standard absorber, which we used to calibrate the spectrum. Our additional studies have shown that a small deviation from this value does not lead to significant changes in the parameters of the partial spectra in our model. However, it is precisely for this value that it is possible to obtain the most satisfactory description of the experimental spectra (the smallest c-square value).

The residual spectrum at 633 K (Fig. 3 a) shows substantial misfit at the peaks. This shows that even a QS distribution is not enough to fit the spectrum. Is it possible that the fixed 0.24 mm/s width is too large for this case?

Our response: In order to have all distributions at the same conditions (number of point of distributions, linewidth, smoothing coefficient,…) we showed a lower quality of the fit to demonstrate the only sharp maximum at the distribution curve. Actually, the spectrum can be fitted better with the use of one doublet function. However, changing the line width did not affect the quality of the description of the spectra. We believe that the main reason for the apparent deviations from the experimental curve is due to smoothing coefficient, used for spectra fitting with distribution reconstruction.

When evaluating spectra with overlapped doublets, it is very important to make sure that there is no texture (preferred orientation of the crystallites of the Mössbauer sample). The Materials and Methods section is very short. The sample preparation should be discussed. The SEM image in the Suppl. shows granular particles (no needles or platelets), thus  the texture is unlikely, but it should be stated in a written form that the texture is excluded in the samples. Otherwise the 2-3 doublets seen in the spectra may be virtual.

Our response: We completely agree with the Reviewer. Texture may be one of the reasons for the spectrum asymmetry. Therefore, in order to exclude this possibility, measurements were carried out at a magic-angle (q » 54.700) between the radiation direction and the normal to the sample plane for some of the samples. No changes have been observed.

This text: „The direction of the lone electron pair is shown, which is opposite to the direction of displacement of the Bi3+ cations” is not very clear to me. The lone pair has no direction itself, it has a position. I would simply say: the Bi3+ center moves/is shifted/is displaced toward the lone pair. (See lines 202, 212,213)

Our response: We completely agree with the Reviewer. The text was rephrased in the manuscript and in the caption.

I recommend publication of this work after minor revisions.

All language corrections were taken into account and were added to the manuscript with gratitude.

Reviewer 2 Report

Comments and Suggestions for Authors

In the submitted manuscript the authors report about 57Fe Mössbauer spectroscopy on multiferroic BiMn7O12. Accompanied are the investigations by synchrotron, SEM, DSC, and magnetic measurements. Main aim is to study the interplay between orbital and spin degrees of freedom mainly by comparison of the electric field gradient with changes in the crystal structure. By changing the temperature structure passes through several crystallographic phases, and changes between para- and ferroelectric behaviour. Comparison with older results obtained for BiMn7O12 show, that 57Fe substitution does not drastically change the local structural, electronic and magnetic properties. The experimental results are compared with theoretical ones. Conclusions are convincing and interesting.

The paper is well organized, discussion is very detailed and clear. To streamline the paper, part of the theoretical calculations are separated in 4 Appendices. A few figures are outsourced into a supplement.

Beside a few corrections/suggestions (see below) the main shortage of the paper is, that almost entirely derivative data are given, but only few hyperfine parameters are reported.   Data are only presented as figures. A table with all the obtained hyperfine parameters (with errors) should be added in the supplement information.

Some comments/suggestions:

Line 139: here it is stated, that “Further lowering of  …. does not affect DSC curves.” Because DSC measurements were restricted to the temperature range 273 to 673 K and transition is expected at 240 K, it is not clear how this statement is justified.

Line 298: the arrow shows in wrong direction.

Line 613: There is no Equation (8S) in the supplement.

Table 1: hyperfine parameters should be explained in caption. Especially D, which is explained much later in the paper.

Figure 5: direction of lone electron pair is opposite to direction of displacement. Why arrows in figure are parallel?

Figure 7: line 360: change 10 K to 101 K

Appendices B and C are not mentioned in the main text.

Reference list must be checked: e.g.:

(a) In the main text 73 papers are mentioned, but in the reference list only 72 are shown.

(b) in line 563 ref [38] is wrong. It should be [37]

(c) in line 564 ref [39] is wrong. It should be [38]

Author Response

In the submitted manuscript the authors report about 57Fe Mössbauer spectroscopy on multiferroic BiMn7O12. Accompanied are the investigations by synchrotron, SEM, DSC, and magnetic measurements. Main aim is to study the interplay between orbital and spin degrees of freedom mainly by comparison of the electric field gradient with changes in the crystal structure. By changing the temperature structure passes through several crystallographic phases, and changes between para- and ferroelectric behaviour. Comparison with older results obtained for BiMn7O12 show, that 57Fe substitution does not drastically change the local structural, electronic and magnetic properties. The experimental results are compared with theoretical ones. Conclusions are convincing and interesting.

The paper is well organized, discussion is very detailed and clear. To streamline the paper, part of the theoretical calculations are separated in 4 Appendices. A few figures are outsourced into a supplement.

Beside a few corrections/suggestions (see below) the main shortage of the paper is, that almost entirely derivative data are given, but only few hyperfine parameters are reported.   Data are only presented as figures. A table with all the obtained hyperfine parameters (with errors) should be added in the supplement information.

Our response: We completely agree with the Reviewer. The tables for each temperature region have been added to SI part.

Some comments/suggestions:

Line 139: here it is stated, that “Further lowering of  …. does not affect DSC curves.” Because DSC measurements were restricted to the temperature range 273 to 673 K and transition is expected at 240 K, it is not clear how this statement is justified.

Our response: DSC data were collected from 125 K, but were not shown for convenience of perception since there were no peaks observed in the range 125 – 300 K.

Line 298: the arrow shows in wrong direction.

Our response: Corrected

Line 613: There is no Equation (8S) in the supplement.

Our response: Corrected

Table 1: hyperfine parameters should be explained in caption. Especially D, which is explained much later in the paper.

Our response: We completely agree with the Reviewer. The information was added to the Table.

Figure 5: direction of lone electron pair is opposite to direction of displacement. Why arrows in figure are parallel?

Our response: We completely agree with the Reviewer. The arrows show the displacements directions, the Figure caption was changed for more clear meaning.

Figure 7: line 360: change 10 K to 101 K

Our response: There is not a misprint. The crystal structure data was taken at 10K and used for the same crystal structure temperature range (below ~270K) at 101K. Below 101K the manganite is in the magnetic ordering state and EFG calculation cannot be compared with quadrupole shift values.

Appendices B and C are not mentioned in the main text.

Our response: Corrected.

Reference list must be checked: e.g.:

(a) In the main text 73 papers are mentioned, but in the reference list only 72 are shown.

(b) in line 563 ref [38] is wrong. It should be [37]

(c) in line 564 ref [39] is wrong. It should be [38]
Our response: Corrected.